# Regulation of RNA Polymerase I Stability and Function

**DOI:** 10.3390/cancers14235776

**Published:** 2022-11-24

**Authors:** Stephanie Pitts, Marikki Laiho

**Affiliations:** 1Department of Radiation Oncology and Molecular Radiation Sciences, Johns Hopkins University School of Medicine, Baltimore, MD 21205, USA; 2Sidney Kimmel Comprehensive Cancer Center, Johns Hopkins University School of Medicine, Baltimore, MD 21205, USA

**Keywords:** RNA polymerase I, enzyme regulation, transcription, ribosome biogenesis, cancer, small molecule therapeutic

## Abstract

**Simple Summary:**

Building ribosomes for cellular protein translation is a massive, energy-consuming undertaking. The process is executed by all cells to replenish the relevant pools of proteins required for cellular functions. Cancer cells are strikingly dependent on this activity, as they need continuous protein synthesis for sustained proliferation and growth. Ribosome biogenesis requires the activities of three RNA polymerases, which transcribe essential RNAs that make up the backbone of the ribosome, and hundreds of proteins, which provide structural and functional support. Of the three RNA polymerases, RNA polymerase I executes a critical and rate-limiting step by transcribing three key ribosomal RNAs. Pol I transcription is pervasively deregulated in cancers, enabling unlimited protein synthesis. Here, we review this enzyme and provide examples of current efforts to target Pol I transcription therapeutically.

**Abstract:**

RNA polymerase I is a highly processive enzyme with fast initiation and elongation rates. The structure of Pol I, with its in-built RNA cleavage ability and incorporation of subunits homologous to transcription factors, enables it to quickly and efficiently synthesize the enormous amount of rRNA required for ribosome biogenesis. Each step of Pol I transcription is carefully controlled. However, cancers have highjacked these control points to switch the enzyme, and its transcription, on permanently. While this provides an exceptional benefit to cancer cells, it also creates a potential cancer therapeutic vulnerability. We review the current research on the regulation of Pol I transcription, and we discuss chemical biology efforts to develop new targeted agents against this process. Lastly, we highlight challenges that have arisen from the introduction of agents with promiscuous mechanisms of action and provide examples of agents with specificity and selectivity against Pol I.

## 1. Introduction: RNA Polymerase I Is an Essential Enzyme for Ribosome Biogenesis

### 1.1. Ribosome Biogenesis

Ribosome biogenesis is a complex and metabolically costly process involving the activities of three cellular RNA polymerases, their RNA products, and hundreds of proteins [1,2,3,4,5]. As protein synthesis is required for cell growth and survival, actively dividing cells produce 1–2 million ribosomes per cell cycle [2]. The process of ribosome biogenesis accounts for over 75% of nuclear transcription in yeast, with 60% of transcription accounting for the transcription of ribosomal RNA (rRNA) and 15% accounting for the transcription of ribosomal proteins [2,6]. Ribosome biogenesis is tightly regulated by metabolic and environmental conditions such as cell growth, nutrient availability, and stress [2,7,8,9]. Dysregulation of ribosome biogenesis is widely implicated in disease, as increased activity is detected in cancer while its defects can lead to ribosomopathy syndromes and developmental disorders (for reviews see [8,10,11]).

The ribosome is a ribonucleoprotein complex responsible for translating mRNA into proteins [4,5]. The eukaryotic 80S ribosome is composed of a small 40S and a large 60S subunits [4,5]. The 40S subunit, consisting of 18S ribosomal rRNA (rRNA) and about 33 ribosomal proteins, binds to messenger RNA (mRNA) and monitors the pairing of an mRNA codon with a transfer RNA (tRNA) anticodon. The 60S subunit, consisting of 5S rRNA, 5.8S rRNA, 28S rRNA, and about 49 ribosomal proteins, catalyzes the peptide bond formation between amino acids [4,5]. Aberrant production of ribosomes underlies the pathophysiology of ribosomopathies and stems from alterations in ribosomal proteins, ribosome biogenesis factors, and defects in RNA polymerase genes (for reviews see [8,10,11,12]).

### 1.2. RNA Polymerases Have Diverged RNA Synthetic Targets

In 1969, Roeder and Rutter made the seminal discovery that eukaryotes possess three different DNA-dependent RNA polymerases (Pols), in contrast to the single RNA polymerase found in prokaryotes [13]. Over the next few decades, the structures and functions of three mammalian polymerases (Pol I, Pol II, and Pol III) were identified. Pol I synthesizes most rRNA, Pol II synthesizes mRNA and other non-coding and regulatory RNAs, whereas Pol III synthesizes tRNA and 5S rRNA [3,14,15,16]. Ribosome biogenesis thus requires the cooperation of all three RNA polymerases. Pol I transcribes a long precursor rRNA which is processed into the mature 5.8S, 18S, and 28S rRNA, Pol II transcribes small nucleolar RNAs (snoRNAs) and the mRNAs required to translate the ribosomal proteins, and Pol III transcribes 5S rRNA [3,14,15,16] (Figure 1). While the catalytic activity of these enzymes is conserved, each enzyme has evolved to transcribe a different set of genes and responds to an intricate set of co-factors and cellular cues [3,15]. Furthermore, while Pol II and Pol III are predominantly nuclear polymerases, Pol I transcription is compartmentalized to the nucleolus.

### 1.3. RNA Polymerase I Transcribes the Essential 5.8S, 18S, and 28S Ribosomal RNAs

The nucleolus forms around the active sites of rRNA gene transcription, which are defined by the multicopy rRNA gene clusters termed as nucleolar organizing regions (NORs) [17]. Human NORs are present on the short arms of acrocentric chromosomes 13, 14, 15, 21, and 22. The number of repeats varies by chromosome between 20–70 and has a substantial interindividual variation, ranging between a total of 100–600 copies in a diploid genome [18,19]. The relevance of this copy number variation on human development, physiology, and pathophysiology is unknown [20,21]. Additionally, the rRNA gene sequences vary from chromosome to chromosome and human to human [18]. This is surprising, given the fundamental nature of the rRNA transcripts. Variant-calling studies using high-coverage whole-genome sequencing data across 26 populations in the 1000 Genomes Project have revealed substantial heterogeneity in the rRNA gene sequences across these populations [22]. Intriguingly, sequence variants are detected in the human-expanded helical folds ES7L and ES27L of the 28S rRNA genes, suggesting that this variation is a late evolutionary addition and has the potential to affect ribosome function [22].

The ribosomal DNA (rDNA) gene repeats are arranged head-to-tail [17,18,23] (Figure 1A). Each rDNA repeat unit consists of a 13 kb coding region and a 30 kb non-coding intergenic spacer (IGS). The Pol I 47S promoter is embedded into the IGS domain of the preceding gene copy. The promoter contains a core promoter and an upstream control element (UCE), which are both obligatory for transcription initiation and are recognized by the pre-initiation complex (PIC) factors in a species-specific manner. Interestingly, a largely duplicated promoter element, termed a “spacer promoter,” exists in humans and mice 1–2 kb upstream of the transcription start site [7,24]. The duplicated promoters are separated by repetitive elements called enhancers, which are essential for the spacer promoter activity [7]. In *Xenopus*, where they were first discovered and are best studied, the enhancers influence the number of activated promoters, but are not absolutely required for basal transcription or polymerase loading [25,26,27,28]. The overall arrangement of the spacer and 47S promoters and enhancers is conserved, albeit their numbers are not. Enhancers stimulate promoters in *cis* and inhibit unlinked promoters in *trans*, but related regulatory factors remain to be identified. We do not know the relevance of these elements and whether they regulate transcription in humans. The spacer promoter is separated from the functional promoter by transcription termination recognition sequences essentially blocking read-through transcription [24]. In mice, the spacer promoter may produce non-coding RNAs that may be involved in gene silencing [29,30].

The first, rate-limiting step of ribosome biogenesis is the transcription of rDNA by Pol I into a 47S rRNA precursor [9,31,32]. The long 47S precursor rRNA is composed of 5’external transcribed spacer (5’ETS), 18S rRNA, internal transcribed spacer 1 (ITS1), 5.8S rRNA, ITS2, 28S rRNA, and 3’ETS and is processed through a series of endonucleolytic and exonucleolytic cleavage steps that eliminate the ETS and ITS sequences [33,34,35] (Figure 1A). Several transcription termination sequences after the 3’ETS terminate Pol I transcription [24,36]. Other than the aforementioned promoters, enhancers, and transcription terminators, the IGS sequences also contain simple repeats, transposable elements, and non-coding sequences that are transcribed by Pol II [18], reviewed in [37].

### 1.4. Pol I Transcription and Early Steps of Ribosome Biogenesis Are Compartmentalized in the Nucleolus

While Pol I machinery is predominantly localized in the fibrillar center of the nucleolus, Pol I transcription occurs on the border of the fibrillar center and dense fibrillar component [23]. The Pol I transcribed 47S precursor rRNA is co-transcriptionally modified and assembled in the dense fibrillar component [6,35]. snoRNAs, transcribed by Pol II in the nucleus, are imported to the nucleolus to induce 2′-O-methylation and the conversion of certain uridines to pseudo-uridines [5,6]. Ribosomal proteins, transcribed by Pol II in the nucleus and translated in the cytoplasm, are imported to the nucleolus to assemble with the mature rRNAs. 5S rRNA, transcribed by Pol III, is imported into the nucleolus as well [4,5,6] (Figure 1B). Over 200 additional ribosome assembly factors participate in the modification and assembly process [4,5]. The 47S precursor rRNA is folded, modified, and assembled to form a 90S precursor particle. This particle is further processed and cleaved into the pre-40S and pre-60S ribosomal subunits. At this time, the 18S rRNA forms the backbone of the pre-40S ribosomal subunit and the 5S, 5.8S, and 28S rRNAs form the backbone of the pre-60S ribosomal subunit. After additional processing, these two pre-subunits are further folded, modified, and assembled in the granular component of the nucleolus [4,5,6]. They are then exported through the nucleus, where further maturation takes place, to the cytoplasm for final maturation into a translation-competent ribosome [4,5] (Figure 1B).

## 2. Pol I Transcription Cycle

### 2.1. Structural Analyses of the Pol I Enzyme in Saccharomyces cerevisiae

Structural studies of Pol I have detailed the enzyme and the preinitiation complex in the various steps in the transcription cycle, namely, preinitiation, initiation, and elongation ([38,39,40,41,42,43,44,45,46,47], reviewed in [3]). The majority of these studies have been conducted in the yeast model *Saccharomyces cerevisiae*. The nomenclature for yeast and human subunits is different and has evolved over time. We refer the reader to a recent review for the naming convention [3] and identify here the key subunits with their current species-specific names. In yeast, Pol I is composed of 14 subunits, with a 10-subunit core that resembles a crab claw [38,39]. The two largest subunits, A190 (human RPA1) and A135 (human RPA2) form the catalytic core and the DNA binding cleft [38,39]. The core encloses a central cleft, where the template DNA binds and the nascent rRNA strand is synthesized. The cleft has two channels, for the entry of substrate nucleoside triphosphates (NTPs) and exit of the RNA product [38,39].

In addition to the catalytic subunits, the 10-subunit core is formed by several subunits shared with Pol II and Pol III (ABC27, ABC23, ABC14.5, ABC10α, ABC10β) and with Pol III (AC40, AC19), which provide mostly structural support for the core assembly [reviewed in [3]. The 10-subunit core of the Pol I complex also contains A12.2 (human RPA12), a subunit composed of two Zn-binding β-ribbon domains. The N-terminal ribbon connects A12.2 with A190, A135, and a heterodimer of A49-A34.5 (human RPA49 and RPA34, respectively) subunits [38,39]. The C-terminal ribbon extends into the active site and interacts with the NTP entry channel and is involved in RNA cleavage during backtracking/proofreading and termination [38,39]. Importantly, the A12.2 C-terminus is only positioned into the active site when Pol I is not in the elongation phase and is displaced during elongation to allow NTP addition [38,39,40,41].

Pol I contains four peripheral subunits unique to this enzyme: A43 (human RPA43), A14, A49, and A34.5. The A43-A14 heterodimer forms the stalk, a structure that protrudes from the core. Preinitiation factor Rrn3 (human RRN3) binds to the A43-A14 stalk and recruits the Rrn3-Pol I complex to the rDNA promoter [40,41,44,45,46,47]. A34.5-A49 are homologous to the Pol II TFIIE/F heterodimer, but in contrast to their auxiliary arrangement in Pol II, are integral to the Pol I complex [48]. They associate with the Pol I lobe (A135) and cavity and A12.2 N-terminal domain. A49 has a tandem winged helix (tWH) domain that forms a bridge over the Pol I DNA binding cleft, which is mobile and displaced to allow promoter DNA binding. The tWH domain affects the enzyme processivity, while the C-terminus is required for DNA binding in vitro and orients towards the upstream promoter [40,41,43]. A49 helps recruit Rrn3-bound Pol I to the promoter and aids in the release of Rrn3 after promoter clearance [41]. Furthermore, the A49-A34.5 heterodimer binds to and stabilizes A12.2 and stimulates its RNA cleavage activity [38,39]. In addition to transcription initiation, the A49-A34.5 heterodimer has been implicated in transcription elongation [49].

### 2.2. Structural Analyses of the Human Pol I Enzyme

Within the past year, two groups have elucidated the structure of human Pol I for the first time [50,51] (Figure 2). Overall, the general structure of Pol I is remarkably conserved between yeast and humans. The main difference is that human Pol I only has 13 subunits, as it does not contain a homolog of the yeast Pol I subunit A14. Instead, the human Pol I RPA43 stalk is introduced with a hinge that is more flexible. This flexibility permits binding to RRN3 while being firmly anchored to the Pol I core [50,51]. Another remarkable difference from yeast Pol I is the wider size of the human Pol I exit tunnel, which accommodates double-stranded RNA, and a larger funnel. These features in particular can support co-transcriptional folding of rRNA and greater elongation speeds [51].

### 2.3. Key Steps in the Pol I Transcription Cycle

Pol I is a highly efficient RNA polymerase that has evolved for the transcription of its long 13 kb transcript. The structure of Pol I, with its in-built RNA cleavage ability and incorporation of subunits homologous to transcription factors, enables it to quickly and efficiently synthesize the enormous amount of rRNA required for ribosome biogenesis [52,53]. This is facilitated by its high initiation rate and elongation speed [54]. Many Pol I complexes are densely packed on the rDNA, with an elongation complex every 140 bp on the actively transcribed genes in yeast [55]. Transcription occurs through tightly regulated cycles [9,31,54].

***Initiation*.** The 140-160-base pair rRNA gene promoter contains two key elements, a core element and the UCE [7,31]. Binding of the mammalian upstream binding factor (UBF) to the UCE and core element mimics a nucleosomal fold. UBF recruits and activates selectivity factor 1 (SL-1) [7,56]. While UBF is present throughout the bodies of actively transcribed coding genes, SL-1 is only present on the gene promoters. SL-1 consists of a TATA-binding protein (TBP) and five additional factors: TAF_I_110, TAF_I_48, TAF_I_63, TAF_I_12, and TAF_I_41 [7,31]. RRN3 interacts tightly with the RPA43 stalk of Pol I [51]. Similarly, the association of RRN3 with RPA43 prevents the enzyme dimerization and maintains Pol I in its monomeric form [44,45,46,47,51]. SL-1 subunits TAF_I_63 and TAF_I_110 interact with RRN3 on the RRN3-Pol I complex and recruit Pol I to the promoter. Together, these proteins form the PIC. This process is relatively conserved in yeast, as yeast has Rrn3, TBP, and a core complex homologous to SL-1 [44,45,46,47]. However, yeast also have an upstream activating factor (UAF), which aids in initiation but has no mammalian homologue [54].

The successful assembly of the PIC bends the promoter to an angle, remarkably different from that of the Pol II promoter, favorable for transcription initiation. Pol I stutters on the first few nucleotides, during which the DNA duplex melts to favor the formation of the transcription bubble and promoter escape. This process is further assisted by A49 and A12.2, which support the binding of Pol I to the promoter and the formation of the open elongation complex, respectively [40,43]. Pol I re-configures from its open complex (bound to DNA) into its elongation complex (synthesizing RNA), followed by dissociation of RRN3 [40,43]. With these features in place, Pol I is positioned for not only the high initiation rates but also the rapid transition into the elongation phase.

***Elongation*.** In the active site, two magnesium cations in the catalytic aspartate triad coordinate an NTP condensation reaction. During this reaction, one nucleoside monophosphate (NMP) is added to the nascent RNA strand as pyrophosphate is released. Pol I then translocates down the DNA template by one nucleotide [54]. This translocation is aided by other elements in the active site, such as the bridge helix, rudder, fork loop(s), trigger loop, and wall [3]. As each NMP is added to the 3′ end of the growing RNA chain, one NMP from the 5′ end goes into the RNA exit tunnel [57]. Pol I nucleotide addition rates are faster than those of Pol II, but this comes with the cost of a less stable enzyme and higher error rates [58]. The mechanic nucleotide addition has been compared to a “ratchet” that drives the forward reaction. However, the Pol I enzyme is not indifferent to its substrate—the sequence context of the substrate, especially the GC-richness, and the co-transcriptional folding of the RNA product both affect the enzyme elongation rate [59,60,61,62].

***Pausing and Backtracking*.** Given the ratchet motion during elongation, the enzyme is prone to backtracking. Furthermore, if the polymerase encounters an obstacle, such as DNA damage or an incorrect nucleotide, it will pause. The polymerase will then backtrack. Since the “gating tyrosine” in the active site remains “open”, the polymerase can backtrack as much as it needs to. The C-terminus of A12.2 inserts into the active site and cleaves the RNA [63,64]. This cleavage activity provides more efficient backtrack recovery for Pol I compared to Pol II. While the C-terminus of A12.2 has the cleavage activity, it is supported in this task by the N-terminal domains, as well as by the A34.5–A49 heterodimer [65]. Inefficient RNA cleavage further leads to proofreading errors [65], and the fidelity of Pol I transcription decreases by 10-fold in the absence of A12.2 [66].

***Termination*.** Transcription termination elements are positioned on two separate sites on the rDNA gene repeat: at the 3′ end of the transcribed region and upstream of the transcription start site. Transcription termination factor I (TTF-I) binds to the termination element at the 3′ end of the transcribed region, bends the DNA, and triggers Pol I to pause [7]. However, given the RNA cleavage activity of the enzyme and co-transcriptional processing, TTF-1 is not essential for the cleavage step but aids in preventing transcription from continuing into the IGS. A12.2 is required for the Pol I release from the DNA template [54]. Given that the promoter and termination sites are closely spatially positioned, the Pol I-RRN3 complex can be recruited back to the same or another UBF and SL-1-bound rDNA promoter to re-engage with the transcription cycle [67,68].

***Pol I Complex Stability.*** Pol I processivity is facilitated by enzyme stability. However, the mechanistic understanding of which factors govern mammalian Pol I assembly, stability, and localization is incomplete. In yeast, the interaction between A190 and A135 is stable [69], and in mammals, RPA1 and RPA2 have long half-lives (>20 h). We previously showed that the stability of RPA1 (yeast A190) is dependent on its binding partner, RPA2 (yeast A135), as silencing of RPA2 reduced RPA1 protein expression and caused its nucleoplasmic translocation [70].

In yeast, A49 stabilizes the expression of A34.5 and is mediated by a protease-sensitive linker domain. Conversely, depletion of A34.5 also destabilizes A49, suggesting that this complex is inherently unstable if either subunit is missing [49]. UV light causes bulky pyrimidine dimer lesions that stall transcription elongation complexes. At cryo-EM resolution these lesions cause RPA1 structural rearrangements that block the enzyme translocation step [71]. In contrast, the A49 domains remain in an open complex configuration, suggesting that the polymerase keeps scanning for initiation even when facing these extensive blocks. A yet unanswered question is whether the A34.5-A49 dimer dissociates from the core complex during the transcription cycle or when facing elongation blocks [72]. Furthermore, kinetic studies on the nucleotide addition rate have suggested that A12.2 acts as an intrinsic destabilizer of Pol I elongation complex in vitro [73].

External factors, such as zinc availability and temperature, also mediate Pol I stability. Zinc depletion has been shown to induce vacuolar proteolysis of Pol I in yeast [74]. Cold temperatures induce the ubiquitination and destabilization of Rpa190, the yeast homolog of mammalian RPA1 [75]. Since the deubiquitylating enzyme Ubp10 affects Rpa190 stability [75], it is possible that Rpa190 is marked for degradation through the ubiquitin proteasome system. This possibility is intriguing, since the largest subunits of both Pol II and Pol III undergo proteasome-mediated degradation when facing transcription challenges. This is discussed in more detail below [76,77,78,79,80,81].

## 3. Regulation of Pol I Transcription Activity

### 3.1. Metabolic and Environmental Conditions Regulate Pol I Transcription

Pol I transcription is regulated by metabolic states and environmental factors, such as cell growth, nutrient availability, stress, and the cell cycle [reviewed in [2,7,8,9]. Conditions that stimulate growth—such as nutrients, growth factors, and a readily available source of energy—activate Pol I transcription. Conversely, conditions that disrupt growth or cellular metabolism—such as nutrient starvation, senescence, and oxidative stress—attenuate Pol I transcription activity [8,9]. Pol I transcription activity is dependent on the cell cycle, as transcription stops during mitosis, recovers during the G_1_ phase, and resumes full capacity during S phase and G_2_ phase [82]. There are two major ways to regulate Pol I transcription. The first is to change the rate of transcription initiation at active rRNA genes. This is typically achieved through reversible post-translational modifications of Pol I PIC factors, and it is amenable to rapid regulatory events. The second occurs by changes in the number of active (versus silent) rRNA genes. Most often this is achieved through epigenetic modification of the rDNA, and it results in stably altered transcriptional states [24,83].

### 3.2. Post-Translational Modification of Transcription Factors

PIC factors are targets of post-translational modifications. These regulatory events affect UBF, SL-1, and RRN3 [82,84], and reviewed in [8,9,31,85]. For example, growth factors that activate the ERK pathway induce UBF phosphorylation by ERK 1/2. This affects the interaction of UBF and the rDNA and increases Pol I transcription [84]. Upon increased cell growth, mTOR and CK2 phosphorylate UBF to increase Pol I transcription. In addition to targeting UBF, these three kinases (ERK, mTOR, and CK2) phosphorylate RRN3 to the same effect [86,87,88]. Acetylation of SL-1 increases Pol I transcription initiation [89]. In contrast, under conditions of cell stress, JNK phosphorylates RRN3, which diminishes the interaction of RRN3 with Pol I and SL-1 and downregulates Pol I transcription [90]. Energy deficits and a high AMP/ATP ratio activate AMPK, which phosphorylates and inactivates RRN3 and causes the demethylation of the rDNA promoter by KDM2A [91,92].

Cell cycle-dependent transcription of Pol I is also achieved by post-translational modifications of the transcription factors. The phosphorylation of SL-1 subunit TAF_I_110 during mitosis prevents SL-1 from interacting with UBF and prevents transcription initiation [82]. SL-1 is dephosphorylated and reactivated at the end of mitosis, but UBF is not dephosphorylated and reactivated until the end of G_1_ phase. Both SL-1 and UBF are fully active during S phase and G_2_ phase [82,93].

Strikingly, Pol I transcription activity and ribosome biogenesis are also under the control of the circadian clock [94,95]. Transcription, as measured by rRNA production and production of ribosome biogenesis and ribosomal proteins, peaks during the day and night cycles, respectively [94,95]. Intriguingly, unassembled rRNAs are polyadenylated and degraded through the exosome [96]. Collectively, these findings emphasize the precise coordination of these major metabolic activities and states. They further highlight the importance of the ability to regulate Pol I transcription in rapid cycles.

### 3.3. Post-Translational Modification of Transcription Factors

Changing the number of active (versus silent) rRNA genes at an rDNA repeat is regulated by epigenetic mechanisms. The rDNA of actively transcribed genes exists in a nucleosome-deplete, “open” euchromatin configuration that is characterized by DNA hypomethylation, H4ac, and H3K4me2 (reviewed in [8,83]). In addition to binding at the rDNA promoter, UBF also binds throughout the rDNA coding region and the IGS. It displaces linker histone H1 and contributes to the decondensed state of the euchromatic rDNA [8,97,98]. The rDNA of silent genes exists in a “closed” heterochromatin state, characterized by H3K9me, H3K20me, and CpG methylation (reviewed in [8,83]). Conditions that support growth also affect rDNA chromatin remodeling and the conversion from the heterochromatin to the euchromatin state. As one mechanism, growth factors which activate PI3K lead to its interaction with and activation of SGK1 kinase. SGK1 recruits histone demethylase KDM4A to nucleolar chromatin. KDM4A interacts with Pol I, binds to the rDNA promoter, demethylates H3K9me3, and upregulates Pol I transcription [99]. In addition, growth factors activate ERK, which phosphorylates UBF. Phosphorylated UBF remodels the rDNA chromatin to promote Pol I transcription elongation [100].

Approximately half of the rRNA genes are maintained in an active state. However, the number of active rRNA genes differs based on cell type, suggesting that the number of active genes is passed down through cell lineages during development and differentiation [83]. This indicates that epigenetic modifications of the rDNA can have long-term effects on the regulation of Pol I transcription. In addition to transcription termination, TTF-I recruits chromatin modifiers to the rDNA such as Cockayne syndrome protein B (CSB) and repressive nucleolar chromatin remodeling complex (NoRC) (reviewed in [8,83]). NoRC acts as a scaffold and recruits DNA methyltransferases, histone deacetylases, and histone methyltransferases to remodel the rDNA to the “closed” heterochromatin state ([101,102,103], reviewed in [8,83]). NoRC also moves the promoter-bound nucleosome further downstream of the transcription start site, which inhibits the formation of the PIC (reviewed in [8,83]). Numerous proteins and small RNAs epigenetically modify the rDNA to influence gene expression (reviewed in [8,83]).

## 4. Transcriptional Errors Evoke Cellular Stress Responses

### 4.1. Pol I Transcription Stress Response

Of all genes, the rRNA genes are the most heavily transcribed. Given the high transcriptional activity and the speed of the polymerase on the long 13 kb transcript, this polymerase is particularly sensitive to transcriptional stresses [35]. A surrogate marker for transcription stress has been a change in the shape and size of the nucleolus, particularly in the altered subcellular localization of many nucleolar proteins [104,105,106,107]. This response has been called the nucleolar stress response, and it is prominently activated by stressors that inhibit rRNA transcription, such as transcription inhibitors or ultraviolet radiation. Pol I transcription stress leads to a structural reorganization of the subnucleolar domains involved in transcription and ribosome biogenesis [107]. rDNA, fibrillar centers, and dense fibrillar components form “caps” around the edges of the nucleoli [107]. Proteins typically found in the nucleolus during ribosome biogenesis, such as ARF, RPL5, and RPL11, are translocated to the nucleus. RPL5 and RPL11 in particular are key factors that bind to MDM2, the E3 ligase responsible for degrading p53, and are essential for p53 stabilization [108,109,110]. Activation of p53 transcription function and its target genes leads to cell cycle arrest and/or activates apoptosis [108,109,110]. These prominent events have also been called the ribotoxic stress response or ribosome biogenesis checkpoint and are reviewed in [12,111,112]. Given the high Pol I transcriptional activity in cancers and the frequency at which transcriptional obstacles are encountered, it is likely that this creates significant pressure particularly in cancers to inactivate p53 by mutation to avoid launching of its tumor suppressive activities.

### 4.2. Pol II and Pol III Transcription Stress Activate Enzyme Destruction

RNA polymerases encounter obstacles generated by altered chromatin conformation or DNA adducts, such as those caused by DNA damage and UV irradiation, that stall or arrest transcription. Each polymerase has its own error rate, with Pol II having the highest fidelity, closely matched by Pol I, while Pol III error rates are the highest [66]. To resolve transcription blockage, Pol II tries to (1) bypass the obstacle or lesion or (2) initiate the transcription-coupled nucleotide excision repair pathway. If neither method resolves the stalling, the cell initiates a “last resort” pathway in which the largest subunit of Pol II, RPB1, is poly-ubiquitinated and degraded by the proteasome [76]. The ubiquitination is mediated by multiple E3 ligases such as Rsp5 (NEDD4L), the elongin ABC/Rbx1/Cul5 complex, BRCA1/BARD1, WWP2, the CRL4 (CUL4, DDB1, RBX1) complex, and CUL3 [113,114,115,116,117,118,119,120,121,122,123,124,125]. Recent studies have found that RPB1 ubiquitination at a single site, K1268, can mediate transcription activity, DNA repair, and RPB1 degradation [77,78,79].

The largest subunit of yeast Pol III, C160, is also ubiquitinated and degraded by the proteasome upon transcription stalling. To induce cell stress, Lesniewska et al. [80] treated yeast cells with transcription inhibitors rapamycin, 6-azauracil, and mycophenolic acid. In a separate experiment, they transferred yeast from fermentation to respiration conditions. These stressors resulted in the proteasome-mediated degradation of C160 [80]. Wang et al. [81] found that defective or stalled Pol III complexes underwent sumoylation of subunit C53, followed by ubiquitination and proteasomal degradation of C160. The ubiquitination was mediated by the Slx5-Slx8 SUMO-targeted E3 ligase complex [81]. Since the stress-induced degradation of the largest subunit is conserved across Pol I, Pol II, and Pol III, this represents an evolutionarily significant means of regulating RNA polymerase activity and resolving the stalled complexes.

## 5. RNA Polymerase I and Cancer

### 5.1. Pol I Transcription Is Upregulated in Cancer

Pol I transcription is a critical step in ribosome biogenesis and protein translation. As such, it is essential, and proportional, to cancer cell growth and a rate-limiting factor for cancer cell proliferation [9,12]. Over a century ago, tumors were observed to contain enlarged, abnormally shaped nucleoli [126]. To match their increased proliferative and biosynthetic activity, cancer cells depend on pervasive, unabated ribosome biogenesis. This is achieved by upregulating the rate-limiting step of ribosome biogenesis, Pol I transcription [9,12]. Increased nucleolar size of cancer cells is correlated with high Pol I transcription and increased proliferation [127,128]. This link is so striking that many pathologists examine nucleolar size when assessing tissue specimens [128,129,130].

Staining of the NOR using silver, which recognizes both nucleolar proteins and RNA, is termed as argyrophilic nucleolar organizing region (AgNOR) staining, and it has been used extensively to assess tumor specimens [129,131]. The number of AgNOR particles has demonstrated prognostic significance in gastrointestinal cancers (colon, rectum, stomach, and liver), urologic cancers (bladder, kidney, and prostate), breast cancer, melanoma, and lung cancers [129,131].

More recently, we introduced specific detection of the 47S rRNA precursor using chromogenic in situ hybridization for visualization and quantification of Pol I transcription activity in human tissue samples [132]. Using this assay, we showed higher Pol I activity in high-grade prostatic intraepithelial neoplasia and prostate cancer than in normal prostate tissue [132]. Similarly, qPCR has been used to measure the expression of the 47S rRNA precursor 5′ ETS region. The levels of 47S rRNA are higher in colorectal and prostate cancers and cervical intraepithelial neoplasia tissue than in control tissues [133,134,135]. 

Several studies have used surrogate markers for Pol I activity, such as the expression of rRNA methyltransferase fibrillarin (FBL) [136,137] and TTF-I. TTF-I RNA levels are higher in tumor tissues than in normal tissues in patients with colorectal cancer and in patients with hepatocellular carcinoma, and higher TTF-I expression correlated with a worse prognosis [138,139]. A broad survey of expression of ribosomal protein transcripts showed their distinct expression compared to healthy tissues, as well as prognostic implications across cancer types [140]. Furthermore, circulating tumor cells from breast cancer patients had distinct subsets of ribosome and protein synthetic signatures that correlated with poor outcomes [141].

### 5.2. Cancer Drivers Promote Deregulated Pol I Transcription

Cancer cells employ several different mechanisms to increase Pol I transcription and ribosome biogenesis [8,9,16,142,143]. Of all oncogenic drivers MYC is perhaps the most powerful inducer of ribosome biogenesis (reviewed in [144,145]). MYC stimulates the expression of Pol I-associated transcription factors, ribosomal, ribosome biogenesis, and other nucleolar proteins, enabling high Pol I transcription rates and increased ribosome biogenesis [136,146,147,148,149]. Nucleophosmin (NPM1), a highly abundant nucleolar protein, interacts with MYC, and directs MYC nucleolar turnover and localization [150]. Cytoplasmic mutants of NPM1, highly prevalent in acute myeloid leukemias, lack this interaction and enhance MYC stabilization [150]. NPM1 also causes the nucleolar retention of C/EBPα, which in turn associates with UBF and SL-1 at the promoter and increases rRNA transcription [151] (Figure 3).

mTOR drives multiple steps in ribosome biogenesis and regulates both transcription of rRNA and mRNA synthesis of ribosomal proteins ([152], reviewed in [153]). Several studies have attempted to pinpoint the regulatory steps in Pol I transcription. These suggest that mTOR regulates PIC factor Rrn3 stability [154], phosphorylates RRN3 and UBF and promotes PIC formation [88,155,156], binds directly to rDNA [157], and regulates rRNA processing [158]. The majority of these studies are inferred consequences of rapamycin treatment; they require further validation using definitive approaches.

RUNX1, a transcription factor frequently altered in myelodysplastic syndromes and leukemias, is required for the sustained high level of rRNA synthesis and ribosomal protein translation [159]. Remarkably, the reduction of ribosome biogenesis provided the hematopoietic cells with increased resistance to genotoxic stress and attenuated p53 pathway activation [159]. Interestingly, ribosome biogenesis is increased during epithelial-mesenchymal transition by a mechanism where the repressive marking by the NoRC complex is released and reciprocally, SNAI1 is recruited to the rRNA gene promoter [160]. This is coincident with the nucleolar recruitment of mTORC2 complex factor RICTOR to the nucleolus [160].

Cancer cells possess constitutively active growth signaling pathways [142,143]. The Ras/MAPK and PI3K pathways activate kinases such as ERK and CK2 and phosphorylate Pol I transcription factors, switching the transcriptional program on permanently ([86,87,88], and reviewed in [8,9,31,85,142,143]). SOD1, superoxide dismutase, drives ribosome biogenesis in Kras-driven lung cancer models in mice [161]. This finding has therapeutic implications as SOD is overexpressed in lung cancers and its pharmacological inhibition demonstrates efficacy in *Kras*-driven lung cancer models. Ect2, a guanine nucleotide exchange factor, binds UBF, recruits Rac and NPM to the Pol I promoter and increases Pol I transcription [162]. Ect2 is essential for *Kras-Trp53* driven lung cancer tumorigenesis in mice [162]. Telomerase has been shown to bind rRNA genes and to increase Pol I transcription during Ras-induced hyperproliferation [163]. Therapeutic intervention using telomerase inhibitor imetelstat in cancer cells reduced Pol I transcription and cell growth [163]. ERBB2 tyrosine kinase, also known as HER2/Neu, and amplified in breast cancers, has been described to associate with rDNA and to increase rRNA synthesis [164].

The activities of cyclin-dependent kinases (CDKs) and cyclins are also deregulated in many cancers due to their mutations, amplifications, and uncontrolled activation. Their deregulation leads also to interference of Pol I transcription by phosphorylation of SL-1 and UBF to maintain constitutively high Pol I transcription rates [8,9,16,142,143,165] (Figure 3).

Additionally, epigenetic modification of the rDNA promoter may contribute to the high Pol I transcription activity in cancer cells. For example, hepatocellular carcinomas are characterized by hypomethylation of the rDNA promoter, leading to a persistently “open” euchromatin state and upregulation of Pol I transcription [166]. rDNA hypomethylation has also been observed in endometrial carcinoma [167]. Cancer cells thus hijack Pol I regulation primarily by the post-translational modification of transcription factors, but also by epigenetic modification of the rDNA, to promote constitutively upregulated Pol I transcription rates.

### 5.3. Inactivation of Tumor Suppressors Leads to Uncontrolled Pol I Activity

Several tumor suppressors, which are mutated and inactivated in cancers, control and attenuate Pol I transcription activity [8,9]. The RB1 protein binds to UBF and causes the dissociation of UBF from the rDNA [168]. p53 and PTEN disrupt SL-1 and impede formation of the pre-initiation complex [169,170]. p53 binds to the SL-1 factors TBP and TAF_I_110 to prevent their interaction and productive initiation [169]. PTEN displaces TBP from the rDNA promoter [170]. BRCA1, known for its regulatory activities in DNA damage repair, cell cycle regulation and transcriptional control, also imposes control over Pol I transcription [171]. It has been described to do so by binding to rDNA and interacting with PIC complex factors SL-1 and UBF [171]. Nucleolar tumor suppressor ARF interacts with UBF to block its phosphorylation, thus affecting Pol I transcription activity [172]. Furthermore, ARF inhibits rRNA processing and augments ribosome export [173]. These activities of ARF augment its major effect to bind to and inhibit NPM and MDM2 [174]. Cellular energy sensor AMPK has been reported to phosphorylate RRN3 and prevent its interaction with SL-1 in energy-starved cells, and alternatively, to reduce transcription by increasing the promoter methylation by KDM2A [91,92]. MYBBP1A, a nucleolar protein and tumor suppressor, also attenuates Pol I transcription and rRNA processing [175]. RUNX2 transcription and morphogenesis factor suppresses Pol I promoter activity together with HDAC1 by binding to UBF and SL-1 [93,176]. DICER is a dsRNA processing enzyme essential for miRNA processing. In *Schizosaccharomyces pombe*, Dicer and other RNA interference mutants are inviable in a manner dependent on Pol I transcription [177]. Deletion of A12.2 rescues the lethality, coupling the Pol I transcription machinery to the RNAi pathway [178]. The overexpression of oncogenes and inactivation of tumor suppressors in cancers is therefore a dominant mechanism by which cancers deregulate Pol I transcription activity [8].

### 5.4. Current Strategies to Target Pol I

Given the essential role of ribosome biogenesis in cancer cell growth and the activated state of Pol I in many cancers, Pol I is a rational target for cancer therapeutics. The high DNA metabolic activity due to the activated transcriptional rates creates unique vulnerabilities at this genetic locus, especially to events that cause transcriptional impediments. Enzyme stability is required for effective transcription elongation. This is especially critical for Pol I, given the high initiation rates and fast processivity over the long precursor transcript. Furthermore, the polymerase density, analyzed in detail in yeast, is very high and renders this enzyme highly vulnerable to perturbations caused by lesions or conformational impediments. In fact, several classes of chemotherapeutic drugs—including alkylating agents, anti-metabolites, antibiotics, and topoisomerase inhibitors—disrupt Pol I transcription [179]. Hence, it is plausible that some of their therapeutic activities are conveyed through the mechanism of inhibition of rRNA synthesis. However, the clinical impact of this mechanism or its contribution has not been evaluated.

Alkylating agents (e.g., cisplatin and oxaliplatin) attach alkyl groups to DNA to form DNA adducts. These adducts trigger a DNA damage response and induce cell cycle arrest and cell death [180]. Cisplatin inhibits Pol I transcription [179]. UBF binds to the cisplatin-DNA adduct and is displaced from the rDNA promoter [181,182]. In contrast to cisplatin, oxaliplatin does not trigger a DNA damage response. It instead inhibits rRNA synthesis and triggers the nucleolar stress response, leading to p53 upregulation and cell death [183]. Oxaliplatin therefore has the potential to be particularly effective against cancers with elevated Pol I activity. In fact, oxaliplatin has shown striking efficacy against colorectal and gastrointestinal cancers [183].

Anti-metabolites (e.g., 5-fluorouracil and methotrexate) mimic cellular metabolites, interfering with enzymatic processes and incorporating into DNA and RNA and interfere with rRNA processing [180]. Methotrexate has been shown to reduce Pol I transcription [179].

Chemotherapeutic agents in the class of antibiotics, such as Actinomycin D, intercalate with DNA and induce DNA damage [184,185,186]. Since Actinomycin D prefers sequences rich in guanine and cytosine, it is mostly found to bind to GC-rich rDNA and inhibits both Pol I and Pol II transcription in a concentration-dependent manner ([179], reviewed in [112]). 

Topoisomerases (TOP) are enzymes that are required to resolve DNA supercoiling and facilitate processes such as transcription, replication, and recombination [187,188]. TOP1 relaxes negative supercoils formed behind the transcription complexes, while TOP2 has evolved to relax the positive supercoils ahead of the complexes. These severe torsional stresses are abundant during Pol I transcription. Not surprisingly, a large number of TOP1 and TOP2 inhibitors have been identified as Pol I inhibitors [179]. The anthracycline class of TOP2 poisons (such as doxorubicin, daunorubicin, epirubicin) form covalent complexes with the DNA and TOP2 and lead to irreparable DNA damage, replication arrest, and cell death [187]. Structurally distinct TOP2 inhibitors etoposide, merbarone, and ellipticines inhibit Pol I, and ellipticines and merbarone have been shown to disrupt Pol I pre-initiation complex formation [189,190,191]. Camptothecin and its derivatives (topotecan, irinotecan) inhibit TOP1 enzymes [179]. However, while these drugs interfere with Pol I transcription, they neither target Pol I specifically nor directly.

G quadruplexes are unusual four-stranded secondary DNA structures that are formed by G-G base pairs via Hoogsteen hydrogen bonding [192]. These structures are found in gene promoters, throughout the rDNA, and at telomeres, and they have been associated with processes such as transcription and replication. As such, G quadruplex targeting has emerged as an approach to inhibit these processes in cancer [192,193]. A naphthalene-diimide derivative was identified to bind to rDNA G quadruplexes and inhibit Pol I transcription [194]. Cylene Pharmaceuticals conducted a screen for small molecules that disrupted nucleolin/rDNA G quadruplexes [193]. The screen identified small molecules CX-3543 and CX-5461 [195,196]. While CX-3543 was found to have bioavailability issues and was unsuccessful in clinical trials, CX-5461 has completed phase 1/2 clinical trials [197]. CX-5461 has been stated to disrupt the interaction between SL-1 and Pol I [196,198]. However, CX-5461 causes DNA damage, is a G quadruplex stabilizer, and acts as a radiosensitizer [199,200,201]. Further studies on its mechanisms of action revealed that CX-5461 is in fact a TOP2 inhibitor and conveys its therapeutic efficacy through this mechanism [200,202,203]. Numerous studies which have used CX-5461 are undermined by the complex and promiscuous mechanisms of action of this molecule, especially when inferred to represent mechanisms related solely to inhibition of Pol I transcription. However, despite these challenges, the drug, when applied with mechanism-based knowledge, such as in triple-negative breast cancers, or in cancers with vulnerabilities dependent on impaired DNA repair, may be clinically useful [193].

We have summarized these drugs in Table 1, which includes agents tested in human and mouse cells and models. Omitted from the table are agents that target rRNA processing, such as 5-FU and nucleotide analogs (purine/pyrimidine synthesis inhibitors); agents that target regulatory or metabolic pathways affecting Pol I transcription (such as mTOR inhibitors, regulators of MYC); metabolic inhibitors (mycophenolic acid, IMPDH inhibitors); and agents that act as broad spectrum transcription and CDK inhibitors (5,6-dichloro-1-b-D-ribofuranosyl-benzimidazole (DRB), triptolide, roscovitine, flavopiridol). Additionally, agents targeting the ribosome and protein translation are not included. Readers are referred to a recent review by Zisi covering these topics [112].

It is evident that many challenges remain in the quest to develop effective Pol I inhibitors. Most of the current drugs induce nonspecific, widespread DNA damage. It would be optimal to test drugs that target Pol I transcription directly. Yet, the mechanisms that govern the stability and regulation of the Pol I enzyme itself remain unclear, making it an elusive enzyme to target. While rRNA synthesis is upregulated in many types of cancer, it is also essential for normal cells. An effective cancer therapeutic must therefore selectively target Pol I transcription to exploit cancer cell vulnerabilities without affecting normal cells. Currently there are only few pharmacological tools that qualify as specific and selective Pol I inhibitors for this purpose.

## 6. Direct Regulation of the Pol I Enzyme to Treat Cancer: BMH-21, A First-in-Class Pol I Inhibitor

We recently discovered a first-in-class small molecule, BMH-21, that specifically and selectively blocks Pol I transcription [206,210]. BMH-21 is unique compared to other chemotherapeutic drugs, such as Actinomycin D, oxaliplatin and topoisomerase II poisons (CX-5461, anthracyclines) in a number of ways. BMH-21 is a DNA intercalator that binds to rDNA non-covalently, does not induce DNA damage [199,206,210,211], and does not act through TOP2 [202,210]. Instead, it induces rapid degradation of RPA1 in a proteasome-dependent manner [206,212]. Using gene silencing, we have shown that the drug efficacy depends on the expression of RPA1, affirming that the molecule acts through the identified target [213]. We and others have described further derivatives of BMH-21 as well as several other small-molecule chemotypes with similar mechanisms of action against Pol I [207,214,215,216,217].

BMH-21 blocks three critical steps in the transcription cycle, namely initiation, promoter escape, and elongation [70,218] (Figure 4). BMH-21 perturbs transcription elongation in vitro and in cells, as ChIP-qPCR shows that the Pol I complex is rapidly disengaged from the rDNA [70,206,218]. Remarkably, these activities are conserved in yeast, facilitating the analysis of its effect on yeast nascent RNA synthesis by NET-seq [70,218]. These analyses revealed that, as predicted by the affinity of BMH-21 to GC-rich sequences, BMH-21 causes Pol I pausing upstream of G-rich sequences [218]. Furthermore, yeast cells expressing elongation-impaired mutants of A190 or A135 are sensitized to BMH-21-mediated loss of viability [70]. Lastly, using kinetic nucleotide addition analyses, Jacobs et al. [219] showed that BMH-21 selectively inhibits Pol I transcription elongation.

RPA1 degradation is prominently observed in cancer cells and is associated with BMH-21-induced cancer cell death [206]. The degradation is only observed in transcription competent cells, linking the polymerase stability to transcriptional activity [70]. The change in RPA1 half-life by BMH-21 is profound, decreasing the half-life to ~1 h, while other Pol I subunits are not affected. RPA1 degradation is mediated by the ubiquitin proteasome system [206], drawing parallels to prior studies on the proteasome-mediated degradation of large RNA polymerase subunits [75,76,77,78,79,80,81]. We have shown that a deubiquitinase (DUB) USP36 inhibits degradation of RPA1 by BMH-21 [206]. Other labs have identified conditions regulating Pol I subunit stability in yeast [74,75]. These studies link the activity of DUBs, such as Ubp10 (a USP36 homolog), to A190. BMH-21 does not affect RPB1, the Pol II catalytic subunit, under conditions in which RPA1 is degraded [206]. On the other hand, cell stresses that cause RPB1 degradation do not affect RPA1 stability [206]. These findings suggest the engagement of different factors in sensing the arrested Pol I and II complexes and affecting their stability. Since the initial discovery of the drug-induced destabilization of RPA1, several other molecules have been shown to increase its turnover [207,208,209] (Table 1).

By conducting an unbiased RNAi screen against ubiquitome proteins, we recently identified SCF^FBXL14^ as an E3 ligase involved in the BMH-21-induced degradation of RPA1 [212]. We showed that knockout and knockdown of FBXL14 abrogated the drug-induced turnover and increased RPA1 half-life. However, the E3 ligase did not affect RPA1 abundance or transcriptional activity in the basal, non-drug inducible state. We also showed that FBXL14 overexpression activated RPA1 turnover in cancer cell lines that were resistant to this degradation [212]. However, RPA1 degradation is not the sole driver of sensitivity to the BMH-21-mediated therapeutic efficacy. While we observed that increased expression of FBXL14 enhanced the sensitivity to BMH-21-mediated cell death in some cancer cell lines, it was not observed in all [212]. We infer that degradation of RPA1 occurs as a consequence of Pol I inhibition, and that the therapeutic activity primarily results from the prominent drug-induced transcription inhibition and ensuing transcriptional and translational stresses (Figure 4). However, the findings suggest that the unsuccessful assembly of Pol I on the rDNA sensitizes RPA1 for rapid turnover and identify FBXL14 as a key mediator of this event. In addition, given that the RNAi screen identified a number of additional E3 ligase candidates affecting both the basal and regulated turnover of RPA1, we predict that a number of E3 ligases will be eventually confirmed to affect the stability of this enzyme in analogy to the large number of E3 ligases affecting RPB1.

The discovery of BMH-21 is exciting for both fundamental and translational reasons. The compound induces the proteasome-mediated degradation of the Pol I catalytic subunit, analogous to prior studies of the proteasome-mediated degradation of the large subunits of Pols II and III. Prior studies focused on the regulation of Pol I activity through the modification of its transcription factors and the rDNA, but these results show that the Pol I enzyme itself can also be regulated through the degradation of its catalytic subunit. This finding thus provides essential knowledge about the stability and regulation of the Pol I complex relevant for the mechanistic understanding of the enzyme function. BMH-21 is the first compound to specifically and selectively inhibit Pol I without having off-target effects, such as those by other chemotherapeutic drugs. This class of small molecules thus has very promising potential in the clinic, and preclinical studies are currently underway on BMH-21 analogs.

## 7. Conclusions

Pol I is responsible for the first and rate-limiting step of ribosome biogenesis—the transcription of rDNA into a 47S rRNA precursor. To match their proliferative and biosynthetic activities, cancer cells possess abnormally high rates of ribosome biogenesis. Since major cancer drivers constitutively activate Pol I transcription, Pol I is a rational target for cancer therapeutics. Current studies focus on the regulation of Pol I activity through the modification of its transcription factors and the rDNA, but very little is known about the stability and regulation of the enzyme itself. The finding that the Pol I complex can be regulated through the degradation of its catalytic subunit provides fundamental knowledge about the stability and regulation of the enzyme relevant for the mechanistic understanding of the enzyme function and the development of Pol I—targeting agents. Preclinical studies have detailed the precise mechanisms of how the BMH-21 class of small molecules inhibits Pol I transcription. Development of these molecules for clinical testing is ongoing. These findings open up a new path for the discovery of other molecules that take advantage of this new regulatory mechanisms.

## Figures and Tables

**Figure 1 cancers-14-05776-f001:**
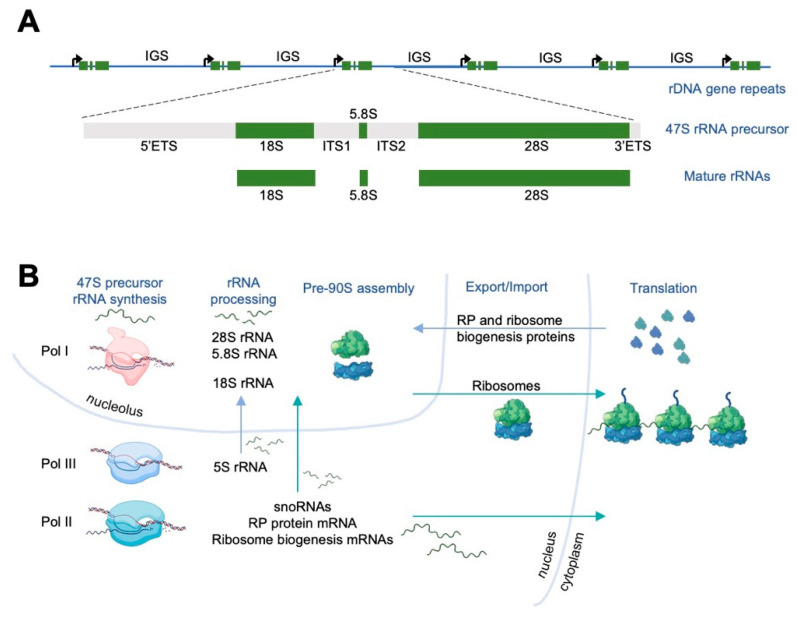
Ribosome biogenesis and activities of RNA polymerases. (**A**) rDNA gene repeats and the Pol I transcribed rRNAs. (**B**) Three polymerases contribute to ribosome biogenesis.

**Figure 2 cancers-14-05776-f002:**
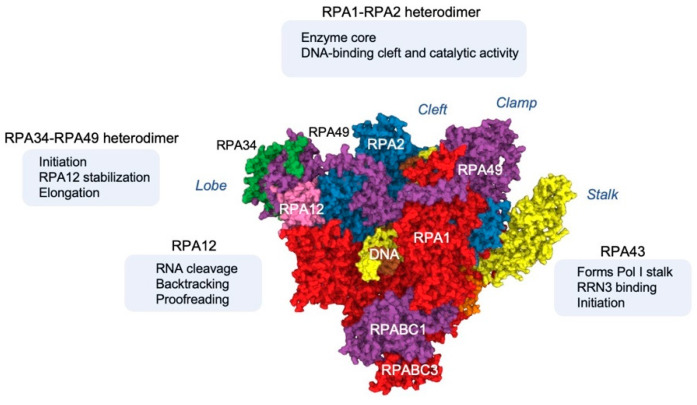
Human Pol I complex subunits. Key elements of the human elongating Pol I complex subunits according to ref. [51]. The complex is rendered from Protein Data Bank structure, PDB:7OB9.

**Figure 3 cancers-14-05776-f003:**
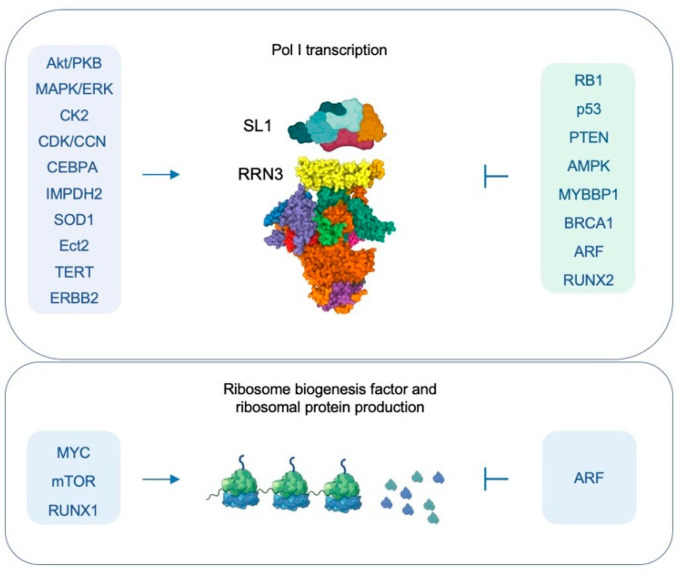
Major effects of positive and negative regulators of Pol I transcription and ribosome biogenesis in cancer. Human Pol I initiation complex depicted from Protein Data Bank structure, PDB:7OBA. Ref. [51].

**Figure 4 cancers-14-05776-f004:**
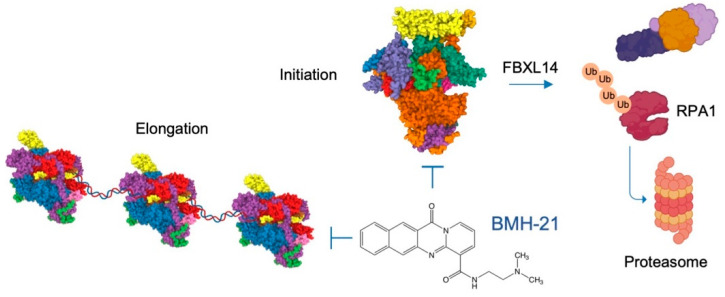
Mechanisms of action of the BMH-21 class of molecules. BMH-21 blocks Pol I transcription initiation, promoter escape, and elongation. This activates consequent destruction of RPA1 mediated by the F-box E3 ligase FBXL14. Human Pol I complexes are depicted from Protein Data Bank structure, PDB:7OB9 and 7OBA. Ref. [51].

**Table 1 cancers-14-05776-t001:** Experimental and clinical agents inhibiting Pol I transcription.

Agent	Drug class	DNA Damage	Clinical/Preclinical	Effect on Pol I Stability	Ref.
Actinomycin D	DNA intercalator	Yes	Clinical	No effect	[179]
Amodiaquine	DNA intercalator	No	Clinical	Destabilization	[204]
9-aminoacridine	DNA intercalator	No	Clinical (topical)	NA *	[205]
BMH-21	DNA intercalator	No	Preclinical	Destabilization	[206]
BMH-9, -22, -23	DNA intercalator	No	Preclinical	Destabilization	[207]
Cisplatin	DNA crosslinker	Yes	Clinical	No effect	[179]
CX-5461	TOP2 inhibitor/G4-stabilizer	Yes	Clinical trials (I/II)	No effect	[199,200,202,203]
CX-3543	G4-stabilizer	NA	Clinical trials (I/II)	No effect	[195]
T5	G4-stabilizer	NA	Preclinical	Destabilization	[194]
Doxorubicin	TOP2 inhibitor	Yes	Clinical	No effect	[179]
Ellipticine	TOP1/2 inhibitor	Yes	Preclinical	NA	[191]
Hernandonine	Alkaloid	No	Preclinical	Destabilization	[208]
Mitoxantrone	TOP2 inhibitor	Yes	Clinical	No effect	[179]
Mitomycin C	DNA intercalator	Yes	Clinical	NA	[179]
Oxaliplatin	DNA crosslinker	Yes	Clinical	No effect	[183]
Sempervirine	Nucleic acid binding	No	Preclinical	Destabilization	[209]
Topotecan	Top1 inhibitor	Yes	Clinical	No effect	[179]

* NA, not available.

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
