# Peer review of "Regulation of RNA Polymerase I Stability and Function"

_cancers, 2022, doi:10.3390/cancers14235776_

Round 1

Reviewer 1 Report

The manuscript ‘Regulation of the RNA Polymerase I Enzyme Stability and Function’ by Pitts and Laiho provides a comprehensive overview of the structural and functional aspects of regulating Pol I activity. 

After introducing ribosome biogenesis, the central importance of nucleolar Pol I transcription for making ribosomes is highlighted. The authors provide detailed insights in the structural aspects of Pol I, describe regulators of Pol I activity, the nucleolar stress response and various modes of deregulating Pol I in cancers. The review closes with an overview of drugs that interfere with Pol I activity. 

The submitted manuscript by Pitts and Laiho is well-written, timely and appropriate for the readers of Cancers. Some minor changes are suggested that may further improve the manuscript: 

-all the figures seem somewhat rudimentary. Given the amount of information in the text, the content of the figures could be designed denser, illustrating more information. For example, in Figure 1 a scheme of the rDNA array and rRNA processing may be beneficial for the non-expert reader. Likewise, the structure in Figure 2 is not very informative. The authors may consider a cycle of structures that represent major steps in the transcription cycle. A lot of space is spent on the regulation of Pol I and its deregulation in cancer, but again, in figure 3 the information content could also be presented in a table. An alternative could be to display 3 conditions side by side: healthy cells, cancer cells and inhibitors of Pol I along with the potential to induce nucleolar stress and impair growth. 

-the part 4 on cellular stress reviews also the impact of stress on Pol II and Pol III in part 4.2. The authors may consider removing this, as it seems superfluous for Pol I functions.

-in part 6 the authors focus mainly of their own previous work on BMH-21. In fact, the bulk of information presented here is based on non-published observations. The authors should refer to a published preprint or merge part 5 and 6, with a reduced focus on BMH-21.

-the part 2 appears somewhat lengthy and not completely in line with parts 1 and 3. After introducing ribosome biogenesis with focus on mammals, most information on the structure of Pol I is based on work in yeast, whereas the mechanisms to regulate Pol I and its deregulation in cancer are presented based on work in mammals again. It may be beneficial to focus on the mammalian aspects in part 2 as well.

-The term ‘Nucleophosmin (NPM)’ should be changed to ‘Nucleophsomin 1 (NPM1)’ to avoid confusion with Nucleoplasmin (NPM2).

Author Response

The manuscript ‘Regulation of the RNA Polymerase I Enzyme Stability and Function’ by Pitts and Laiho provides a comprehensive overview of the structural and functional aspects of regulating Pol I activity. 

After introducing ribosome biogenesis, the central importance of nucleolar Pol I transcription for making ribosomes is highlighted. The authors provide detailed insights in the structural aspects of Pol I, describe regulators of Pol I activity, the nucleolar stress response and various modes of deregulating Pol I in cancers. The review closes with an overview of drugs that interfere with Pol I activity. 

We thank very much for this expert review and have edited the text for further clarity and precision.

The submitted manuscript by Pitts and Laiho is well-written, timely and appropriate for the readers of Cancers. Some minor changes are suggested that may further improve the manuscript: 

-all the figures seem somewhat rudimentary. Given the amount of information in the text, the content of the figures could be designed denser, illustrating more information. For example, in Figure 1 a scheme of the rDNA array and rRNA processing may be beneficial for the non-expert reader. Likewise, the structure in Figure 2 is not very informative. The authors may consider a cycle of structures that represent major steps in the transcription cycle. A lot of space is spent on the regulation of Pol I and its deregulation in cancer, but again, in figure 3 the information content could also be presented in a table. An alternative could be to display 3 conditions side by side: healthy cells, cancer cells and inhibitors of Pol I along with the potential to induce nucleolar stress and impair growth. 

Thank you for these suggestions. We have incorporated suggestions for Figure 1.

-the part 4 on cellular stress reviews also the impact of stress on Pol II and Pol III in part 4.2. The authors may consider removing this, as it seems superfluous for Pol I functions.

We consider the comparisons between the three polymerases relevant and have retained the chapter.

-in part 6 the authors focus mainly of their own previous work on BMH-21. In fact, the bulk of information presented here is based on non-published observations. The authors should refer to a published preprint or merge part 5 and 6, with a reduced focus on BMH-21.

All statements on BMH-21 are referenced and based on published original articles.

-the part 2 appears somewhat lengthy and not completely in line with parts 1 and 3. After introducing ribosome biogenesis with focus on mammals, most information on the structure of Pol I is based on work in yeast, whereas the mechanisms to regulate Pol I and its deregulation in cancer are presented based on work in mammals again. It may be beneficial to focus on the mammalian aspects in part 2 as well.

Thank you for this comment. We have provided further clarity on description of findings described in yeast and mammalian models and have improved the text by new subheadings in Section 2.

-The term ‘Nucleophosmin (NPM)’ should be changed to ‘Nucleophsomin 1 (NPM1)’ to avoid confusion with Nucleoplasmin (NPM2).

We have corrected the reference of NPM to NPM1.

Reviewer 2 Report

Review of the manuscript Cancers-2045885

The manuscript entitled “Regulation of the RNA Polymerase I Enzyme Stability and Function” by Stephanie Pitts and Marikki Laiho, review the current research on the regulation of Pol I transcription and the chemical biology efforts to develop new targeted agents against this process in a context of cancer care.

The review starts by introducing the central involvement of RNA Polymerase I (Pol I) in ribogenesis through the production of most ribosomal RNAs (rRNAs). Pol I activity constitutes the first step of ribosome biogenesis: transcription of rDNA by Pol I into a 47S rRNA precursor (in human). The 47S precursor rRNA is folded, modified, assembled and cleaved into the pre-40S and pre-60S ribosomal subunits After additional processing, these two pre-subunits are further folded, modified, assembled and are then exported through the nucleus to the cytoplasm for final maturation into a translation-competent ribosome. The authors highlighted the central importance of accurate ribosomal biogenesis to carry out healthy cell cycle. Indeed, dysregulation of ribosome biogenesis is widely implicated in disease, such as cancer and developmental disorders.

In a second part, Pol I transcription cycle and the various elements regulating its activity, in a normal or stressed context, are detailed. A focus is made on Pol I structure and its conserved characteristics with Pol II and Pol III. However, some discrepancies are highlighted, such as the presence of built-in transcription factors and the incomplete understanding of the involvement of all subunits on the elongation complex stability and accuracy upon elongation and/or pauses and backtrack events.

In addition, the authors reviewed known way of Pol I transcriptional activity regulation. The two describe pathway to regulate Pol I transcription is to change the rate of transcription initiation at active rRNA genes, or by changing the number of active (versus silent) rRNA genes. Approximately half of the rRNA genes are maintained in an active state. However, the number of active rRNA genes differs based on cell type, suggesting that the number of active genes is passed down through cell lineages during development and differentiation. Moreover, cell cycle-dependent transcription of Pol I is also achieved by post-translational modifications of the transcription factors. Interestingly, Pol I transcription activity and ribosome biogenesis are also under the control of the circadian clock. Indeed, transcriptional activity, as measured by rRNA production and ribosomal proteins production, peak during the day and night cycles, respectively.

Finally, the last third of the manuscript is focused on the deregulation of Pol I in cancer cells and the current strategy to target Pol I in cancer care, notably through the use of BMH-21, a first-in-class Pol I inhibitor. First of all, authors highlighted that Pol I is essential, and proportional, to cancer cell growth. Moreover, they listed papers using Pol I activity markers, such as the accumulation of the 47S rRNA precursor (probe with 5’-ETS probe), that suggest their differential expression across various cancer types compared to healthy tissues. In addition, they listed several different mechanisms employed by cancer cells to increase Pol I transcription and ribosome biogenesis, such as the inactivation of tumor suppressors leading to uncontrolled Pol I activity. From these results, scientific community started to consider Pol I as rational target for cancer therapeutics. Most of the agent inhibiting Pol I, directly or indirectly, described in the review and that are engaged in clinical or preclinical trials displays DNA damage, such as CX-5461, a TOP2 inhibitor. Fortunately, some of them, such as BMH-21, a DNA intercalator, seem to be more promising. It specifically and selectively inhibits Pol I without having off-target effects such as those by other chemotherapeutic drugs. This class of small molecules thus has very promising potential in the clinic, and preclinical studies are currently underway on BMH-21 analogs.

Altogether, this review clearly demonstrates that Pol I is a rational target for cancer therapeutics, since major cancer drivers constitutively activate Pol I transcription. Moreover, Pol I transcription regulation and stability of the elongation complex, which still need to be understood, appears to be fundamentally relevant in the strategy of using targeting agent against Pol I in cancer therapeutics. Indeed, this review propose a list of agents displaying antitumor effect by affecting Pol I activity. A focus is made on a particularly promising drugs called BMH-21, which induces downregulation of Pol I activity by promoting degradation of the elongation complex.

Major comments:

Authors should clearly mention that this review is focusing on human Pol I. Without this precision in the beginning of the review, some part are not accurate. Studies in yeast do not need validation. However, to generalize results obtained in budding or fission yeast, used a model system of eukaryotic cells, studies in metazoan, and/or human cells, are needed. This is not always clarified. Most mechanistic studies are performed in yeast, and not in human. Some parts of the text are difficult to read, with an introduction on yeast, and a conclusion in human. See page 1, line 39 (number of ribosomes being produced) vs page 2 line 44 (cancer and diseases). Similar confusion are present in the part 5.2 (Page 10, line 439 to 442), authors mention studies performed in yeast… require further validation using definitive approaches. UBF, or MYC, is not present in yeast and this part is not clear as it is.

Nomenclature of Pol I subunit is important in this review, and authors should clarify this with little more details (Page 4, line 146 ). Identification of three nuclear RNA polymerase (in the yeast Saccharomyces cerevisiae) were called A, B, C. by Sentenac lab. After extensive purification, Sentenac lab identified polypeptide associated with Pol I (A), by their apparent molecular weight (A190, A34.5, A49, A12.2, ect…). Polypeptides common to Pol I and III are named AC40 and AC19. Polypeptides that are common to the three Pols are ABC27, ABC23, ABC14.5, ABC10alpha and ABC10beta.

Budding yeast nomenclature is now three letters, followed by a number. This nomenclature is now been used for RNA Pol II, with all polypeptides named “Rpb” for RNA polymerase B, and a number (1 to 12). As state in this review, all common polypeptides are called rpb (ABC27 is Rpb5, ect…). The two polypeptide shared between Pol I and Pol III are re-named Rpc40 and Rpc19. In budding yeast Pol I, genetic studies describe polypeptides as Rpa190, Rpa135, Rpa49, Rpa34, Rpa14 and Rpa12. Such nomenclature should be clearly mentioned to allow readers to identify gene (RPA190, RPA135, RPA49, RPA34, RPA14 and RPA12) or polypeptides (A190, A135, A49, A34.5, A14 and A12.2…) in published studies. This should be clearly mentioned clarified. One error in nomenclature should be corrected: for example A34.5 (page 4, line 158) and A34 (page 6, line 235, line 255, 256, 262).

Similarly, Human Pol I subunits should not be only described as Rpa49/Rpa34. Human Pol I subunits orthologs to Rpa49 was initially cloned by Muramatsu lab, and named PAF53 (for polymerase I associated factor of 53kDa). Orthologs of Rpa34 was either called PAF49, or CAST, and only later Rpa34.

Minor comments:

Page 2, line 59: A reference is missing. Two studies identified three eukaryotic RNA polymerases, Roeder work is mentioned, but not Kedinger et al., (PMID: 4907405) with the identification of α-amanitin. The definition of three nuclear Pol (A,B,C or I, II, III) are built on those two studies.

Page 3, line 105: authors claims that rRNA production by Pol I is rate limiting for ribosome biogenesis. This is an important information, but I am not sure that this is completely established. In some specific example, transcription of rDNA by Pol I cannot be considered as a rate-limiting step of ribosome biogenesis. In budding yeast, an overproduction of rRNA do not promote an increase of ribosome biogenesis, as described in Darrière et al., (2020).

Page 3, line 113: A transition should be appreciated between the paragraph dealing with the rRNA resulting from Pol I transcription and the following paragraph explaining the localization of Pol I in the nucleolus.

Page 5, lines 212-213: The transition from PIC, and productive initiation is not fully accurate. Pol I, associated with Rrn3, most likely transcribes approximately 1kb (in human) before Rrn3 dissociation from the elongation complex (see Herdman et al., PLOS genetics, 2017- PMID: 28715449).

Page 6, line 224 : The reference 58 is relevant, but reference 66 could have been added here (Gout et al ;, Sci Adv 2017).

Page 6, lines 229 to 237: This paragraph suggest that a pause or a backtrack is necessarily a consequence of an obstacle encountered. However, it is important to note that RNA polymerases translocation is based on Brownian ratchet motion, making elongation prone to frequent backtracking and potentially sensitive to quite modest forces (Dangkulwanich et al., 2013 for Pol II. PMID: 24066225). Co-transcriptional folding of the nascent rRNA has been proposed to prevente backtracking, thereby favoring productive elongation (ref 6 : Turowski et al., 2020).

Page 9, line 412: Measuring 5’ETS amount levels by qPCR gives information about accumulation of all pre-rRNA species containing 5’ETS sequences. Therefore, 47S accumulation will be measured, among other shorter rRNA species. This is an indirect measure to evaluate Pol I transcriptional activity. The formulation “expression of 47S” should be replaced by “accumulation of 47S, revealed by 5’ETS detection”.

Page 11, line 499-500: This statement should be more precise. Yeast can be fission yeast (Schizosaccharomyces pombe) or budding yeast (Saccharomyces cerevisiae). Reference 177 explore the role of dicer in quiescence of fission yeast. Reduction of rRNA expression allows fission yeast to survive to the absence of dicer (suppressor screen). Dicer is not existing in budding yeast. Reference Roche et al., RNA biol. , 2017 PMID: 28497998 could also be mentioned.

Page 12, lines 542-543: Not clear as it is, and the original reference is missing. The use of the terms “preceding” and “following” is confusing as it is. It should be replaced by another formulation, similar to the original publication Lui and Wang, PNAS, PMID: 2823250, “the DNA in front of the polymerase becomes overwound, or positively supercoiled; the DNA behind the polymerase becomes underwound, or negatively supercoiled.”

Page 14, line 602 and 636, (Pitts 2022, under revision) is now accepted, and should now be mentioned as PMID: 36372232 (congratulations!).

Author Response

The manuscript entitled “Regulation of the RNA Polymerase I Enzyme Stability and Function” by Stephanie Pitts and Marikki Laiho, review the current research on the regulation of Pol I transcription and the chemical biology efforts to develop new targeted agents against this process in a context of cancer care.

The review starts by introducing the central involvement of RNA Polymerase I (Pol I) in ribogenesis through the production of most ribosomal RNAs (rRNAs). Pol I activity constitutes the first step of ribosome biogenesis: transcription of rDNA by Pol I into a 47S rRNA precursor (in human). The 47S precursor rRNA is folded, modified, assembled and cleaved into the pre-40S and pre-60S ribosomal subunits After additional processing, these two pre-subunits are further folded, modified, assembled and are then exported through the nucleus to the cytoplasm for final maturation into a translation-competent ribosome. The authors highlighted the central importance of accurate ribosomal biogenesis to carry out healthy cell cycle. Indeed, dysregulation of ribosome biogenesis is widely implicated in disease, such as cancer and developmental disorders.

In a second part, Pol I transcription cycle and the various elements regulating its activity, in a normal or stressed context, are detailed. A focus is made on Pol I structure and its conserved characteristics with Pol II and Pol III. However, some discrepancies are highlighted, such as the presence of built-in transcription factors and the incomplete understanding of the involvement of all subunits on the elongation complex stability and accuracy upon elongation and/or pauses and backtrack events.

In addition, the authors reviewed known way of Pol I transcriptional activity regulation. The two describe pathway to regulate Pol I transcription is to change the rate of transcription initiation at active rRNA genes, or by changing the number of active (versus silent) rRNA genes. Approximately half of the rRNA genes are maintained in an active state. However, the number of active rRNA genes differs based on cell type, suggesting that the number of active genes is passed down through cell lineages during development and differentiation. Moreover, cell cycle-dependent transcription of Pol I is also achieved by post-translational modifications of the transcription factors. Interestingly, Pol I transcription activity and ribosome biogenesis are also under the control of the circadian clock. Indeed, transcriptional activity, as measured by rRNA production and ribosomal proteins production, peak during the day and night cycles, respectively.

Finally, the last third of the manuscript is focused on the deregulation of Pol I in cancer cells and the current strategy to target Pol I in cancer care, notably through the use of BMH-21, a first-in-class Pol I inhibitor. First of all, authors highlighted that Pol I is essential, and proportional, to cancer cell growth. Moreover, they listed papers using Pol I activity markers, such as the accumulation of the 47S rRNA precursor (probe with 5’-ETS probe), that suggest their differential expression across various cancer types compared to healthy tissues. In addition, they listed several different mechanisms employed by cancer cells to increase Pol I transcription and ribosome biogenesis, such as the inactivation of tumor suppressors leading to uncontrolled Pol I activity. From these results, scientific community started to consider Pol I as rational target for cancer therapeutics. Most of the agent inhibiting Pol I, directly or indirectly, described in the review and that are engaged in clinical or preclinical trials displays DNA damage, such as CX-5461, a TOP2 inhibitor. Fortunately, some of them, such as BMH-21, a DNA intercalator, seem to be more promising. It specifically and selectively inhibits Pol I without having off-target effects such as those by other chemotherapeutic drugs. This class of small molecules thus has very promising potential in the clinic, and preclinical studies are currently underway on BMH-21 analogs.

Altogether, this review clearly demonstrates that Pol I is a rational target for cancer therapeutics, since major cancer drivers constitutively activate Pol I transcription. Moreover, Pol I transcription regulation and stability of the elongation complex, which still need to be understood, appears to be fundamentally relevant in the strategy of using targeting agent against Pol I in cancer therapeutics. Indeed, this review propose a list of agents displaying antitumor effect by affecting Pol I activity. A focus is made on a particularly promising drugs called BMH-21, which induces downregulation of Pol I activity by promoting degradation of the elongation complex.

We thank very much for this expert review and have edited the text for further clarity and precision.

Major comments:

Authors should clearly mention that this review is focusing on human Pol I. Without this precision in the beginning of the review, some part are not accurate. Studies in yeast do not need validation. However, to generalize results obtained in budding or fission yeast, used a model system of eukaryotic cells, studies in metazoan, and/or human cells, are needed. This is not always clarified. Most mechanistic studies are performed in yeast, and not in human. Some parts of the text are difficult to read, with an introduction on yeast, and a conclusion in human. See page 1, line 39 (number of ribosomes being produced) vs page 2 line 44 (cancer and diseases). Similar confusion are present in the part 5.2 (Page 10, line 439 to 442), authors mention studies performed in yeast… require further validation using definitive approaches. UBF, or MYC, is not present in yeast and this part is not clear as it is.

Thank you for these comments. We have provided further clarity and accuracy and edited the text in response to these comments.

Nomenclature of Pol I subunit is important in this review, and authors should clarify this with little more details (Page 4, line 146 ). Identification of three nuclear RNA polymerase (in the yeast Saccharomyces cerevisiae) were called A, B, C. by Sentenac lab. After extensive purification, Sentenac lab identified polypeptide associated with Pol I (A), by their apparent molecular weight (A190, A34.5, A49, A12.2, ect…). Polypeptides common to Pol I and III are named AC40 and AC19. Polypeptides that are common to the three Pols are ABC27, ABC23, ABC14.5, ABC10alpha and ABC10beta.

Budding yeast nomenclature is now three letters, followed by a number. This nomenclature is now been used for RNA Pol II, with all polypeptides named “Rpb” for RNA polymerase B, and a number (1 to 12). As state in this review, all common polypeptides are called rpb (ABC27 is Rpb5, ect…). The two polypeptide shared between Pol I and Pol III are re-named Rpc40 and Rpc19. In budding yeast Pol I, genetic studies describe polypeptides as Rpa190, Rpa135, Rpa49, Rpa34, Rpa14 and Rpa12. Such nomenclature should be clearly mentioned to allow readers to identify gene (RPA190, RPA135, RPA49, RPA34, RPA14 and RPA12) or polypeptides (A190, A135, A49, A34.5, A14 and A12.2…) in published studies. This should be clearly mentioned clarified. One error in nomenclature should be corrected: for example A34.5 (page 4, line 158) and A34 (page 6, line 235, line 255, 256, 262).

Similarly, Human Pol I subunits should not be only described as Rpa49/Rpa34. Human Pol I subunits orthologs to Rpa49 was initially cloned by Muramatsu lab, and named PAF53 (for polymerase I associated factor of 53kDa). Orthologs of Rpa34 was either called PAF49, or CAST, and only later Rpa34.

We have carefully verified that all subunits are correctly labelled and used the most current yeast/human nomenclature throughout.

Minor comments:

Page 2, line 59: A reference is missing. Two studies identified three eukaryotic RNA polymerases, Roeder work is mentioned, but not Kedinger et al., (PMID: 4907405) with the identification of α-amanitin. The definition of three nuclear Pol (A,B,C or I, II, III) are built on those two studies.

We have retained the first published reference to the polymerases.

Page 3, line 105: authors claims that rRNA production by Pol I is rate limiting for ribosome biogenesis. This is an important information, but I am not sure that this is completely established. In some specific example, transcription of rDNA by Pol I cannot be considered as a rate-limiting step of ribosome biogenesis. In budding yeast, an overproduction of rRNA do not promote an increase of ribosome biogenesis, as described in Darrière et al., (2020).

While we do not disagree with the reviewer that further studies could enlighten this aspect, the notion of the relevance of Pol I transcription on ribosome biogenesis currently still stands.

Page 3, line 113: A transition should be appreciated between the paragraph dealing with the rRNA resulting from Pol I transcription and the following paragraph explaining the localization of Pol I in the nucleolus.

Thank you for this comment. We have provided further subheadings to Section 1 that much improve the flow.

Page 5, lines 212-213: The transition from PIC, and productive initiation is not fully accurate. Pol I, associated with Rrn3, most likely transcribes approximately 1kb (in human) before Rrn3 dissociation from the elongation complex (see Herdman et al., PLOS genetics, 2017- PMID: 28715449).

Thank you. We have corrected the statement accordingly.

Page 6, line 224 : The reference 58 is relevant, but reference 66 could have been added here (Gout et al ;, Sci Adv 2017).

We have kept reference 58 as it most accurately describes the statement made.

Page 6, lines 229 to 237: This paragraph suggest that a pause or a backtrack is necessarily a consequence of an obstacle encountered. However, it is important to note that RNA polymerases translocation is based on Brownian ratchet motion, making elongation prone to frequent backtracking and potentially sensitive to quite modest forces (Dangkulwanich et al., 2013 for Pol II. PMID: 24066225). Co-transcriptional folding of the nascent rRNA has been proposed to prevente backtracking, thereby favoring productive elongation (ref 6 : Turowski et al., 2020).

Thank you. We have corrected the statement accordingly.

Page 9, line 412: Measuring 5’ETS amount levels by qPCR gives information about accumulation of all pre-rRNA species containing 5’ETS sequences. Therefore, 47S accumulation will be measured, among other shorter rRNA species. This is an indirect measure to evaluate Pol I transcriptional activity. The formulation “expression of 47S” should be replaced by “accumulation of 47S, revealed by 5’ETS detection”.

We have clarified the statement.

Page 11, line 499-500: This statement should be more precise. Yeast can be fission yeast (Schizosaccharomyces pombe) or budding yeast (Saccharomyces cerevisiae). Reference 177 explore the role of dicer in quiescence of fission yeast. Reduction of rRNA expression allows fission yeast to survive to the absence of dicer (suppressor screen). Dicer is not existing in budding yeast. Reference Roche et al., RNA biol. , 2017 PMID: 28497998 could also be mentioned.

Thank you. We have clarified the yeast species used in the Dicer studies.

Page 12, lines 542-543: Not clear as it is, and the original reference is missing. The use of the terms “preceding” and “following” is confusing as it is. It should be replaced by another formulation, similar to the original publication Lui and Wang, PNAS, PMID: 2823250, “the DNA in front of the polymerase becomes overwound, or positively supercoiled; the DNA behind the polymerase becomes underwound, or negatively supercoiled.”

Thank you. We have clarified the statement.

Page 14, line 602 and 636, (Pitts 2022, under revision) is now accepted, and should now be mentioned as PMID: 36372232 (congratulations!).

Thank you! The reference has been added.

Reviewer 3 Report

This is a nicely organized and written review summarizing most of what is known about Pol, by authors Pitts and Laihoo. Pol I has been neglected in contrast to the more extensive literature concerning Pol II turnover in response to DNA lesions.

After reading, what I miss is perhaps a more expanded perspective/outlook, on novel and remaining important questions in the field although it is indicated here and there, as for example in the discussion on the identification of novel E3 ligases. There are a few unresolved issues that a reader may contemplate about, also non-specialist when reading. As an example, why do some compounds destabilize Pol I and not others despite highly efficiency inhibition Actinomycin D, does BMH-21 and related compounds produce a special type of rDNA lesion? Pitts and Laihoo write that some aspects of Pol I are “incompletely understood”, and as reflection to this statement, one may even say that some chapters have not been written yet. The above is general feedback, that I’m not asking the authors to correct since I find the manuscript in good shape overall.

I have some minor points that the authors may want to address or reflect upon to improve the manuscript further.

1.     Jacobs RQ 2022, Schneider lab, has now several papers, including a recent in iScience 2022, it could be good to clarify this further and update text accordingly. As I understood, it will be a paper also in this review special issue and it is not the iScience paper? Line 615-617.

2.     Much work on FBXL14 is cited as Pitts 2022 under revision, and this is in my opinion ok, and I leave it up to the authors to decide how much they want to discuss this and may well depend on the progress of the revision. Anyways as I note, the paper can be accessed as a sneak peek on CellPress but requires registration. Given this, I think it is fine to discuss and introduce it.

3.     While BMH-21 has emerged as clearly the most potent compound among a few others in triggering rapid Pol I degradation some statements by Pitts and Laiho are rather strong, but again, I leave it to the authors. Line 661-662 as example. A reader may interpret some statements as BMH-21 directly binds to RPA1 to inhibits it catalytic function but this may not have been shown, please comment and it could be good to be more  clear on this.

4.     It would be interesting for the reader, to get a more detailed and expanded comment on BMH-21 and analogues as regarding future pre-clinical and clinical development (although this compound is not the center topic of the review). Line 663-664.

5.     A few other articles in a recent flurry of interesting ones, that the authors may find of help to include and comment on in Pol II section are: RNA Pol II targeted degradation … Yuanjun LI et al, Mol Cell 82: 3943-3959 (2022), Steurer B et al PNAS, 2018; Jacobs RQ, iScience Nov 18, 2022. And a recent one in Journal Cell Biol on live analysis of Pol I.

6.     It turns out that the nomenclature for Pol I subunits in humans, mice and yeast is complex, unfortunately, but authors correctly use RPA1 for the protein of POLR1A gene, although it is a bit confusing since the gene name for replication protein A1 is RPA1. The authors may want to include all possible older names in brackets in the section line 136 onwards. But perhaps this will be even more confusing for the reader?

7.     The authors do cite papers on Myc and transcription of rDNA by Pol I ( e g Arabi et al, Grandori et al) but they do not have Myc in the figure 3 as a Pol I transcription regulator. Please comment.

8.     Figure 4 is not so clear what the reader can see in the picture besides BMH-21 inhibiting the Pol I protein complex. The inhibition of the various  Pol I transcription activities by BMH-21, is it possible to re-design the figure a bit, so it is visualized differently for improved clarity.

Author Response

This is a nicely organized and written review summarizing most of what is known about Pol, by authors Pitts and Laihoo. Pol I has been neglected in contrast to the more extensive literature concerning Pol II turnover in response to DNA lesions.

After reading, what I miss is perhaps a more expanded perspective/outlook, on novel and remaining important questions in the field although it is indicated here and there, as for example in the discussion on the identification of novel E3 ligases. There are a few unresolved issues that a reader may contemplate about, also non-specialist when reading. As an example, why do some compounds destabilize Pol I and not others despite highly efficiency inhibition Actinomycin D, does BMH-21 and related compounds produce a special type of rDNA lesion? Pitts and Laihoo write that some aspects of Pol I are “incompletely understood”, and as reflection to this statement, one may even say that some chapters have not been written yet. The above is general feedback, that I’m not asking the authors to correct since I find the manuscript in good shape overall.

We thank very much for this expert review and have edited the text for further clarity and precision.

I have some minor points that the authors may want to address or reflect upon to improve the manuscript further.

  1. Jacobs RQ 2022, Schneider lab, has now several papers, including a recent in iScience 2022, it could be good to clarify this further and update text accordingly. As I understood, it will be a paper also in this review special issue and it is not the iScience paper? Line 615-617.

The reference for Jacobs et al, Cancers 2022 has been added to the list of references.

  1. Much work on FBXL14 is cited as Pitts 2022 under revision, and this is in my opinion ok, and I leave it up to the authors to decide how much they want to discuss this and may well depend on the progress of the revision. Anyways as I note, the paper can be accessed as a sneak peek on CellPress but requires registration. Given this, I think it is fine to discuss and introduce it.

The reference for Pitts et al, JBC 2022 has been added to the list of references.

  1. While BMH-21 has emerged as clearly the most potent compound among a few others in triggering rapid Pol I degradation some statements by Pitts and Laiho are rather strong, but again, I leave it to the authors. Line 661-662 as example. A reader may interpret some statements as BMH-21 directly binds to RPA1 to inhibits it catalytic function but this may not have been shown, please comment and it could be good to be more  clear on this.

Thank you for these comments. We have further verified that all statements on the mechanisms of action of BMH-21 are based on published studies.

  1. It would be interesting for the reader, to get a more detailed and expanded comment on BMH-21 and analogues as regarding future pre-clinical and clinical development (although this compound is not the center topic of the review). Line 663-664.

Thank you for these comments. We have added a note on the clinical development to Conclusions.

  1. A few other articles in a recent flurry of interesting ones, that the authors may find of help to include and comment on in Pol II section are: RNA Pol II targeted degradation … Yuanjun LI et al, Mol Cell 82: 3943-3959 (2022), Steurer B et al PNAS, 2018; Jacobs RQ, iScience Nov 18, 2022. And a recent one in Journal Cell Biol on live analysis of Pol I.

We agree with the reviewer on these studies being interesting, but as they relate to Pol II we have not expanded on them in the context of this review.

  1. It turns out that the nomenclature for Pol I subunits in humans, mice and yeast is complex, unfortunately, but authors correctly use RPA1 for the protein of POLR1A gene, although it is a bit confusing since the gene name for replication protein A1 is RPA1. The authors may want to include all possible older names in brackets in the section line 136 onwards. But perhaps this will be even more confusing for the reader?

We have carefully verified that all subunits are correctly labelled and used the most current yeast/human nomenclature throughout. To help the reader, we have provided the corresponding yeast/human names for the key subunits when they are first cited.

  1. The authors do cite papers on Myc and transcription of rDNA by Pol I ( e g Arabi et al, Grandori et al) but they do not have Myc in the figure 3 as a Pol I transcription regulator. Please comment.

We have provided full description of the activities of MYC on ribosome biogenesis in the text. The purpose of the figure is to highlight the key impact of the regulators on ribosome biogenesis.

  1. Figure 4 is not so clear what the reader can see in the picture besides BMH-21 inhibiting the Pol I protein complex. The inhibition of the various  Pol I transcription activities by BMH-21, is it possible to re-design the figure a bit, so it is visualized differently for improved clarity.

We feel the figure appropriately summarizes how BMH-21 interferes with Pol I transcription.